# Isolation and Characterization of Neutralizing Monoclonal Antibodies from a Large Panel of Murine Antibodies against RBD of the SARS-CoV-2 Spike Protein

**DOI:** 10.3390/antib13010005

**Published:** 2024-01-05

**Authors:** Emanuela D’Acunto, Alessia Muzi, Silvia Marchese, Lorena Donnici, Valerio Chiarini, Federica Bucci, Emiliano Pavoni, Fabiana Fosca Ferrara, Manuela Cappelletti, Roberto Arriga, Silvia Maria Serrao, Valentina Peluzzi, Eugenia Principato, Mirco Compagnone, Eleonora Pinto, Laura Luberto, Daniela Stoppoloni, Armin Lahm, Rüdiger Groß, Alina Seidel, Lukas Wettstein, Jan Münch, Andrew Goodhead, Judicael Parisot, Raffaele De Francesco, Gennaro Ciliberto, Emanuele Marra, Luigi Aurisicchio, Giuseppe Roscilli

**Affiliations:** 1Takis Biotech, 00128 Rome, Italy; muzi@takisbiotech.it (A.M.); bucci@takisbiotech.it (F.B.); pavoni@takisbiotech.it (E.P.); ferrara@takisbiotech.it (F.F.F.); cappelletti@takisbiotech.it (M.C.); arriga@takisbiotech.it (R.A.); serrao@takisbiotech.it (S.M.S.); peluzzi@takisbiotech.it (V.P.); principato@takisbiotech.it (E.P.); pinto@takisbiotech.it (E.P.); luberto@takisbiotech.it (L.L.); stoppoloni@takisbiotech.it (D.S.); armin.lahm@gmail.com (A.L.); marra@takisbiotech.it (E.M.); aurisicchio@takisbiotech.it (L.A.); 2INGM-Istituto Nazionale di Genetica Molecolare “Romeo ed Erica Invernizzi”, 20122 Milan, Italy; marchese@ingm.org (S.M.); donnici@ingm.org (L.D.); defrancesco@ingm.org (R.D.F.); 3Department of Pharmacological and Biomolecular Sciences (DiSFeB), University of Milan, 20133 Milan, Italy; 4Evvivax Biotech, 00128 Rome, Italy; chiarini@takisbiotech.it (V.C.); compagnone@takisbiotech.it (M.C.); 5Department of Engineering, Università Campus Bio-Medico di Roma, 00128 Rome, Italy; 6Department of Experimental and Clinical Medicine, University Magna Graecia, 88100 Catanzaro, Italy; 7Institute of Molecular Virology, Ulm University Medical Center, 89081 Ulm, Germany; ruediger.gross@uni-ulm.de (R.G.); alina.seidel@uni-ulm.de (A.S.); jan.muench@uni-ulm.de (J.M.); 8Carterra, 825 N. 300 W., Suite C309, Salt Lake City, UT 84103, USA; agoodhead@carterra-bio.com (A.G.); jparisot@carterra-bio.com (J.P.); 9Tumor Immunology and Immunotherapy Unit, IRCSS Regina Elena National Cancer Institute, 00144 Rome, Italy; gennaro.ciliberto@ifo.it

**Keywords:** SARS-CoV-2, neutralizing antibodies, mAb panel, pancoronavirus, pandemic preparedness, betacoronaviruses

## Abstract

The COVID-19 pandemic, once a global crisis, is now largely under control, a testament to the extraordinary global efforts involving vaccination and public health measures. However, the relentless evolution of SARS-CoV-2, leading to the emergence of new variants, continues to underscore the importance of remaining vigilant and adaptable. Monoclonal antibodies (mAbs) have stood out as a powerful and immediate therapeutic response to COVID-19. Despite the success of mAbs, the evolution of SARS-CoV-2 continues to pose challenges and the available antibodies are no longer effective. New variants require the ongoing development of effective antibodies. In the present study, we describe the generation and characterization of neutralizing mAbs against the receptor-binding domain (RBD) of the SARS-CoV-2 spike protein by combining plasmid DNA and recombinant protein vaccination. By integrating genetic immunization for rapid antibody production and the potent immune stimulation enabled by protein vaccination, we produced a rich pool of antibodies, each with unique binding and neutralizing specificities, tested with the ELISA, BLI and FACS assays and the pseudovirus assay, respectively. Here, we present a panel of mAbs effective against the SARS-CoV-2 variants up to Omicron BA.1 and BA.5, with the flexibility to target emerging variants. This approach ensures the preparedness principle is in place to address SARS-CoV-2 actual and future infections.

## 1. Introduction

Coronavirus disease 2019 (COVID-19) was declared a pandemic in March 2020 by the World Health Organization [1] and, since December 2019, has spread around the world with over 770,563,467 cases and more than 6,957,216 confirmed deaths as of September 2023 (https://covid19.who.int/ accessed on 6 September 2023). Despite the extraordinary efforts made by the entire scientific community regarding vaccine development, COVID-19 has continued to spread globally over the past year with the recurrent emergence of new variants that are highly capable of immune escape [2]. From this perspective, it is essential to develop, in parallel with vaccines, other therapeutic tools capable of rapidly combating up-coming future waves of SARS-CoV-2 infections. MAbs have proved to be very effective, well tolerated and more immediate in regard to administration than other types of antiviral treatments [3]. To date, there are about 100 research and development programs for monoclonal antibodies (mAbs) and small molecules against COVID-19, with involvement by some 200 companies and institutions in around 50 countries [4]. Antibodies developed to recognize viral surface proteins have been studied to be used against infectious diseases, such as HIV, Ebola [5,6] and Middle East respiratory syndrome (MERS) [7,8,9]. During the outbreak of the first severe acute respiratory syndrome coronavirus (SARS-CoV) and MERS-CoV, plasma from convalescent patients was used as an effective treatment to reduce the viral load and mortality [10,11]. In the current COVID-19 pandemic, similarly, a small number of patients treated with plasma from convalescent patients have shown evident clinical improvement and a decrease in viral load [12]. The effort in combating SARS-CoV-2 infection has generated a great number of reported mAbs, with over 20 currently progressing through clinical trials. Even before the onset of the COVID-19 pandemic, a repertoire of over 100 approved mAbs was available for diverse human diseases, including a subset designed to address viral infections such as Palivizumab for respiratory syncytial virus and Ansuvimab for Ebola virus. Several mAbs, and combinations thereof, have obtained regulatory endorsement or emergency use authorization for tackling SARS-CoV-2. These interventions primarily target the early stages of mild-to-moderate COVID-19 cases [13]. Most of the currently characterized anti-SARS-CoV-2 antibodies have been isolated from single memory B cells derived from convalescent patients or from transgenic animals immunized against the virus. Regeneron Pharmaceuticals Inc. (Regeneron), using both convalescent patient-derived B cells and immunized animals, developed two potent antibodies administered as a REGN-COV2 cocktail (REGN10933/casirivimab+REGN10987/Imdevimab) [14], obtaining emergency use authorization (EUA) from the U.S. in November 2020. Similarly, Eli Lilly in collaboration with AbCellera developed LY-CoV555 (also called LY3819253 or Bamlanivimab) [15], also authorized in November 2020, for administration as a single intravenous dose in patients with mild-to-moderate COVID-19. Later on, a high-throughput screening approach was adopted using peripheral blood mononuclear cells (PBMCs) that were isolated from a COVID-19 convalescent donor to identify LY-CoV1404 or Bebtelovimab, which recognized an epitope different from Bamlanivimab [16]. In February 2022, emergency use authorization (EUA) was granted to Bebtelovimab for the management of mild-to-moderate COVID-19 in both adults and pediatric individuals who test positive for COVID-19 and are at a heightened risk of developing severe COVID-19. The collaboration between GSK and Vir Biotechnology, on the other hand, led to the development of VIR-7831/GSK4182136 or Sotrovimab. Similarly, there is AstraZeneca’s Evusheld, a combination of two mAbs (AZD8895/Tixagevimab + AZD1061/Cilgavimab) that earned the FDA’s approval in December 2021 following positive results from its PROVENT trial [17]. SARS-CoV-2 has acquired mutations capable of substantially changing its pathogenicity, providing the concrete evidence that mutations could affect the rate of transmission or the antigenicity of the virus [2]. Given the molecular diversity and ongoing evolution of SARS-CoV-2, many mAbs have lost their recommendation status, necessitating continuous evaluations of their effectiveness [18]. The Food and Drug Administration (FDA) revoked the emergency use authorizations (EUA) both for Bamlanivimab, due to the increasing circulation of SARS-CoV-2 variants, and for Sotrovimab, due to an increased prevalence of resistant Omicron BA.2 sub-variants. Currently, only Bebtelovimab has been documented as retaining its effectiveness against all the SARS-CoV-2 variants considered here [19,20], but the future emergence of newly immune-evasive mutations that also reduce the efficacy of this mAb are likely to occur. 

In the effort to address the evolving landscape of viral variants, we have adopted a powerful approach that bridges genetic and protein vaccination techniques. This strategy seamlessly combines the rapid antibody production facilitated by genetic immunization with the potent immune stimulation offered by protein vaccination, resulting in the effective generation of a diverse array of antibody variants. 

Genetic vaccines, particularly the intramuscular electro-gene-transfer of plasmid DNA (DNA-EGT), have emerged as a safe and efficient method to elicit robust immune responses against a wide range of antigens. DNA-EGT enhances DNA uptake and protein expression in skeletal muscle cells, inducing local inflammatory responses that contribute to the development of strong immune reactions against the target antigen(s) [21].

The integration of genetic and protein vaccination not only enhances the antibody diversity, but also accelerates the development of therapeutic tools against challenging and evolving viruses like SARS-CoV-2. This multiplicity forms the solid base of our comprehensive strategy for neutralizing not only existing viral variants, but also those poised to emerge on the horizon.

As the relentless emergence of new variants poses an ever-growing challenge, we recognize the urgency of developing mAbs adept at effectively targeting and neutralizing these viral adaptations. This study introduces a compelling panel of mAbs engineered to hinder the interaction between the angiotensin-converting enzyme 2 (ACE2) receptor and the receptor-binding domain (RBD) of the SARS-CoV-2 spike protein. These antibodies not only enrich our arsenal of therapeutic tools, but also open avenues to isolate and produce potent antibodies against novel variants. Our core research initiative revolves around the construction of a diverse antibody library, ingeniously combining genetic and classical immunization strategies. From this reservoir, we have successfully isolated two antibodies proficient in combating SARS-CoV-2 variants, up to Omicron variants BA.1 and BA.5. Of paramount significance is the readiness with which we can swiftly isolate additional antibodies to counteract emerging variants. These pivotal findings signify a significant leap forward in the realm of therapeutic tools against SARS-CoV-2 infections, ensuring that we remain agile and adaptive in the face of an ever-evolving adversary.

## 2. Materials and Methods

### 2.1. Protein Production and Purification

RBD-hFc, RBD-6xHis proteins and ACE2 were produced by transient transfection of Expi293F high-density cells (Expi293 ™ Expression System Kit, A14635, Thermo Fisher Scientific, Waltham, MA, USA) with the ExpiFectamine 293 cationic lipid transfection reagent, according to the manufacturer’s instructions. The supernatant containing the proteins was collected one week later and clarified by centrifugation and filtration for the subsequent purification steps. The RBD-hFc and ACE2-hFc proteins were purified by affinity chromatography using the Akta Pure system with a protein A column (ToyoScreen AF-R Protein A HC-650F; Tosoh Bioscience, South San Francisco, CA, USA). Briefly, the column was equilibrated with a binding buffer (sodium phosphate buffer 0.1 M, pH 8.0) and loaded with the supernatant, diluted 1:1 with the same buffer. After washing the column, the protein was recovered by acid elution in 0.1 M, pH 3.0 citrate buffer, neutralized in Tris-HCl, pH 9.0 and subjected to dialysis in phosphate buffered saline (PBS) 1X with a Slide-A-Lyzer, as indicated in the product datasheet. The RBD-6xHis protein and its mutants were purified using affinity chromatography of the His tag residues for metals immobilized by the Akta Pure system with a HisPur™ Ni-NTA chromatography column (Thermo Fisher, Waltham, MA, USA), according to the manufacturer’s instructions. Briefly, the column was equilibrated in PBS1X/Imidazole 5 mM and loaded with the supernatant, diluted 1:1 with the same buffer. After washing, the protein was eluted with PBS 1X/Imidazole 0.3 M, pH 7.4. Finally, the protein was dialyzed against PBS 1X with a Slide-A-Lyzer (Thermo), following the manufacturer’s instructions. Once recovered from dialysis, the RBD-hFc and RBD-6xHis proteins were quantified using a spectrophotometer by absorbance at 280 nm. The purity of the proteins was evaluated using the SDS-PAGE method and western blot analysis, carried out in both reduced and non-reduced conditions, using the 6xHis HRP conjugated antibody and the anti-Fc HRP conjugated antibody for the RBD-6xHis and RBD-hFc proteins, respectively.

### 2.2. Genetic and Protein Vaccination

For the immunization plan, a combined genetic and protein vaccination was adopted, where the genetic vaccination approach was based on plasmid delivery and electroporation directly in mice skeletal muscles [22]. A full-length (FL) cDNA encoding wild-type spike was used for immunization, injecting a plasmid vector containing a codon-optimized cDNA version expressing the WT SARS-CoV-2 spike. The immunization protocol consisted of 2 injections into the quadriceps of the mice with 50 µg of pTK1A-COVID-FL, alternated with 2 intraperitoneal injections of 10 µg with the WT RBD-hFc protein. More precisely, the mice were immunized on days 0 and 14 by genetic immunization with the pTK1A-COVID-FL plasmid, and on days 7 and 21 with the purified protein RBD-hFc. The DNA was formulated in PBS at a concentration of 1 mg/mL, and the DNA-EP was performed with a Cliniporator electroporator and flat electrodes (IGEA, Carpi, Italy), with the following electrical conditions in the electro-gene-transfer (EGT) mode: 8 pulses of 20 msec at 110 V, 8 Hz, 120 msec pause between impulses. The RBD-hFc protein was formulated in PBS/the Sigma Adjuvant System (1:1) (S6322, Merck, Rahway, NJ, USA). Two weeks after the fourth immunization, the mice with the highest antibody titers were euthanized and the spleens and lymph nodes harvested to be used in the somatic fusion process.

### 2.3. Fusion Process

P3X63Ag8653 myeloma cells were cultured in 10% FCS RPMI (Gibco #21875-034, Gibco, Grand Island, NY, USA) supplemented with glutamine (Sigma Aldrich #G7513, Sigma Aldrich, St. Louis, MO, USA) and penicillin/streptomycin (Gibco #15070-063). The spleen cells isolated from selected immunized BALB/c mice were extracted by flushing the spleen with sterile PBS 1X and then crushing the tissue on a 70 µm cell strainer. The splenocytes and myeloma cells were mixed at 2:1 ratio in serum-free media and the fusion process was performed with PEG 1500, using the standard procedure [23]. The cells were then resuspended in the culture medium, and unfused myeloma cells were eliminated by adding the HAT supplement 1X (Gibco #21060-017), starting from the following day. On the fifth day, the cells were plated in 384-well plates and grown in an incubator at 37 °C, 5% CO_2_. The hybridomas were then screened using the antigen RBD ELISA, and the positive cells were picked and transferred into 24-well plates and then fed with the growth medium + HT supplement 0.1 mM (Gibco #11067-030). The positivity of the hybridomas was checked twice using ELISA and then the cells were frozen in a freezing medium (90% FBS/10% DMSO (Sigma-Aldrich #D2650)).

### 2.4. Monoclonal Antibody Production 

The hybridomas of interest, upon confirmation of their positivity, were gradually adapted after thawing and transferring from the medium containing 10% FBS to the CD hybrid medium (Gibco); they were grown and amplified at their maximum density in static until a final volume of about 400 mL was reached. The supernatant containing the mAbs produced by these hybridomas was collected and clarified by centrifugation and filtration for the subsequent purification steps. The mAbs were purified by affinity chromatography using the Akta Pure system with a protein A column (ToyoScreen AF-RProtein A HC-650F; Tosoh Bioscience). Briefly, the column was equilibrated with a binding buffer (sodium phosphate buffer 0.1 M, pH 8.0) and loaded with the supernatant, diluted 1:1 with the same buffer. After washing the column, the protein was recovered by acid elution in 0.1 M citrate buffer pH 3.0, properly neutralized in Tris-HCl, pH 9.0 and subjected to dialysis in PBS 1X with a Slide-A-Lyzer, according to the product datasheet. Selected chimeric and humanized antibodies were produced by transient transfection of the ExpiCHO high-density cells (ExpiCHO™ Expression System Kit, A29133) with the ExpiFectamine lipid cationic transfection reagent, according to the manufacturer’s instructions. The supernatant containing the proteins was collected one week later and clarified by centrifugation and filtration for the subsequent protein A purification steps, as explained in the previous paragraph. The purity of the antibodies was assessed using the SDS-PAGE method and western blot analysis, conducted under both reduced and unreduced conditions and using standard methods, and their ability to bind recombinant RBD was assessed using ELISA.

### 2.5. Affinity Measurement and Competition Evaluation for the ACE2 Receptor Using Surface Plasmon Resonance

The affinity (K_D_) for RBD and the k_a_/k_d_ values were determined using surface plasmon resonance using a Carterra LSA instrument (Carterra, Salt Lake City, UT, USA). A biosensor chip HC200M (Carterra) was covalently immobilized with a goat anti-mouse IgG Fc (Jackson ImmunoResearch #115-005-071). The chip was activated with a 7 min injection of a freshly prepared 1:1:1 (*v*/*v*/*v*) mixture of 400 mM EDC+100 mM N-hydroxysulfosuccinimide (SNHS) + 100 mM 2-(N-morpholino) ethanesulfonic acid (MES) at pH 5.5. Then, the goat anti-mouse IgG Fc was diluted to 75 μg/mL in 10 mM sodium acetate at pH 4.5 and coupled for 15 min. Excess reactive esters were blocked with a 7 min injection of 1 M ethanolamine HCl at pH 8.5. Anti-RBD mAb samples were prepared by diluting the cell culture supernatants 1:4 in HBSTE buffer (10 mM HEPES, pH 7.4, 150 mM NaCl, 3 mM EDTA, 0.05% Tween) and captured onto individual spots for 15 min. The recombinant purified RBD protein was prepared at concentrations of 0, 0.5, 1.4, 4.1, 12.3, 37, 111, 333 and 1000 nM, and these samples were injected as analyte for 5 min, allowing 15 min dissociation time. The samples were injected in ascending concentration, without any regeneration in between. The binding data from the local reference spots (inter-spots, representing the naked capture reagent) were subtracted from the active spots and the nearest buffer blank analyte responses were subtracted to double reference the data. The double-referenced data were fit globally to a simple 1:1 Langmuir binding model using Carterra’s kinetic tool, allowing each spot its own k_a_, k_d_ and R_max_ value. To evaluate these antibodies’ ability to block ACE2 binding to RBD, a 250 nM injection of ACE2 was performed after the last injection of RBD. The recorded binding signals determined whether the antibody competed with ACE2 for the same binding region on the RBD. 

### 2.6. Biolayer Interferometry for Quantitation Assay

A rapid quantitation of the monoclonal antibody in the hybridoma medium was performed by biolayer interferometry (BLI) using the Octet RED96 system (ForteBio, Fremont, CA, USA). Analyses were performed using anti-mouse or anti-human Fc biosensors at 30 °C, with shaking at 1000 rpm in a 96-well plate (microplate 96 well, F-bottom, black, 655209, from Greiner Bio-One), containing 200 µL of solution in each well. The samples were diluted 1:10 in a diluent buffer (sample diluent 18-1104, ForteBio) to reduce medium interference. Moreover, biosensor tips were soaked for 10 min in running buffer, prepared by diluting the hybridoma medium 1:10 in the sample diluent. Quantifications were made taking into consideration the initial values (0 to 180 s) of the binding responses. A calibration curve was prepared using a purified reference antibody diluted in running buffer (concentration range: 0.195–25 µg/mL)

The sample concentration was computed from the standard curve using the Octet Software V11.1. To evaluate the calibration curve goodness-of-fit, the residual (%) of each calibrator was estimated, which resulted in a residual lower than 18% and the r2 of the curve fit used to determine the binding rate was >0.98.

### 2.7. HEK293TN-hACE2 Cell Line Generation 

The HEK293TN-hACE2 cell line was generated by lentiviral transduction of HEK293TN cells. The HEK293TN cells were obtained from System Bioscience, Palo Alto, CA, USA. The lentiviral vectors were produced following a standard procedure, based on calcium phosphate co-transfection with 3rd generation helper and transfer plasmids. The following helper vectors were used (gifts from Didier Trono): pMD2.G/VSV-G (Addgene #12259, Watertown, MA, USA), pRSV-Rev (Addgene #12253), pMDLg/pRRE (Addgene #12251). The transfer vector pLENTI_hACE2_HygR was obtained by cloning hACE2 from pcDNA3.1-hACE2 (a gift from Fang Li, Addgene #145033) into pLenti-CMV-GFP-Hygro (a gift from Eric Campeau and Paul Kaufman, Addgene #17446). The hACE2 cDNA was amplified by PCR and inserted under the CMV promoter of pLenti-CMV-GFP-Hygro after GFP excision with XbaI and SalI digestion, namely pLENTI_hACE2_HygR has been made available through the Addgene (Addgene #155296). After transduction with the hACE2 lentiviral vector, the cells were subjected to antibiotic selection with hygromycin at 250 μg/mL. The expression of hACE2 was confirmed by flow cytometry staining using the anti-hACE2 primary antibody (AF933, R&D Systems) as follows: 400,000 cells were incubated for 1 h at 4 °C using 1µg of the primary antibody in PBS (J61196-AP, Thermo Scientific), 2% FBS. Afterwards, the cells were stained using the rabbit anti-goat IgG secondary antibody (Alexa Fluor 647) diluted 1:200 in PBS (J61196-AP, Thermo Scientific), 2% FBS for 30 min at 4 °C. The HEK293TN-hACE2 cells were maintained in DMEM, supplemented with 10% fetal bovine serum (FBS, Gibco Life Technologies, Frederick, MD, USA), 1% glutamine (L-glutamine, 100X, EuroClone, Pero, Italy), 1% penicillin–streptomycin (P/S, 100X, EuroClone), 1% sodium pyruvate (Na pyruvate, Gibco Life Technologies), 1% non-essential amino acids (NEAA, Gibco Life Technologies) and 250 μg/mL hygromycin (Gibco), and the expression of hACE2 was found to be stable after multiple passages.

### 2.8. Production of SARS-CoV-2 Pseudoparticles 

Pseudoparticles production and pseudovirus assays were carried out in two different ways for different antibody subsets. To generate VSV-based pseudoparticles, 15 × 10^6^ HEK293T cells per flask were seeded into T175 flasks in 20–30 mL DMEM supplemented with 10% FBS, 1% L-glutamine and 1% penicillin–streptomycin. One day after, the cells were transfected with 44 µg of pCG1-SARS-2-S per flask (kindly provided by Stefan Pöhlmann, Infection Biology Unit, German Primate Center, Göttingen, Germany) using TransIT LT-1 (Mirus). After 24 h, the cells were transduced with a replication-deficient vesicular stomatitis virus (VSV) vector, in which the genetic information for its native glycoprotein (VSV-G) was replaced by genes encoding the enhanced green fluorescent protein and firefly luciferase (FLuc) (kindly provided by Gert Zimmer, Institute of Virology and Immunology, Mittelhäusern, Switzerland), and incubated at 37 °C for 2 h. Then, the inoculum was removed, the cells were washed with PBS and a fresh medium was added. After 16–18 h, the supernatants were collected and cleared by centrifugation (2000× *g*, 10 min, RT). In order to block the remaining VSV-G carrying particles, an anti-VSV-G antibody (produced in I1 hybridoma cells; ATCC no. CRL-2700) was added and incubated at room temperature for 5 min. The samples were then aliquoted and stored at −80 °C.

To generate lentiviral particles pseudotyped with the SARS-CoV-2 spike, we constructed a series of expression plasmids each encoding a SARS-CoV-2 spike variant. Briefly, for each variant, the corresponding C-terminal deleted (19 amino acids) spike cDNA was cloned in pcDNA3.1. The pLenti CMV-GFP-TAV2A-LUC Hygro was generated from pLenti CMV GFP Hygro (Addgene #17446) through adding T2A-Luciferase using PCR cloning. To produce the pseudotyped lentiviral particles, 5 × 10^6^ HEK-293TN cells were plated in a 15-cm dish in complete DMEM medium and co-transfected using calcium phosphate-mediated transient transfection on the following day with 32 µg of plasmid pLenti CMV-GFP-TAV2A-LUC Hygro, 12.5 µg of pMDLg/pRRE (Addgene #12251), 6.25 µg of pRSV-Rev (Addgene #12253) and 9 µg of spike plasmid. Additionally, 12 h before transfection, the medium was replaced with complete ISCOVE complemented with 10% fetal bovine serum (FBS, Gibco Life Technologies), 1% glutamine (L-glutamine, 100X, EuroClone), 1% penicillin–streptomycin (P/S, 100X, EuroClone), 1% sodium pyruvate (Na pyruvate, Gibco Life Technologies) and 1% non-essential amino acids (NEAA, Gibco Life Technologies). Then, 30 h after transfection, the supernatant was collected, clarified by filtration through 0.45 μm pore size membranes and concentrated by centrifugation for 2 h at 20,000 rpm using a SW32Ti rotor. Viral pseudoparticle suspensions were aliquoted and stored at −80 °C.

### 2.9. Neutralization Assay

For the neutralization assay with VSV-based pseudoparticles, Vero E6 cells were plated at 6000 cells/well in 96-well plates in 180 µL DMEM containing 2.5% FBS, 1% L-glutamine and 1% penicillin–streptomycin. After 24 h, the hybridoma supernatants or heat-inactivated (56 °C, 30 min) sera were serially titrated in PBS as indicated, mixed with pseudovirus stocks (1:1, *v*/*v*), incubated at 37 °C for 30 min and added to the cells in duplicates (20 µL per well). After 16–18 h of incubation, the transduction efficiency was analyzed. For this, the supernatants were removed, and the cells were lysed with a cell culture lysis reagent (Promega, Madison, WI, USA) (40 µL per well). The lysates were transferred into white 96-well plates, and the luciferase activity was measured using a commercially available substrate (Luciferase Assay System, Promega) and a plate luminometer (Orion II microplate luminometer, Berthold). For the analysis of the raw values (RLU/s), the background signal of the untreated cells was subtracted, and the values normalized to the pseudovirus preincubated with PBS only. The IC50 values of the hybridoma samples were calculated using non-linear regression ([inhibitor] vs. normalized response variable slope) in GraphPad Prism.

To obtain a neutralization assay with lentiviral particles, HEK293TN-hACE2 cells were plated at 10,000 cells/well in white 96-well plates in complete DMEM medium. After 24 h, the cells were transduced with 0.1 MOI of SARS-CoV-2 pseudovirus, previously incubated with a serial 3-fold dilution of antibodies, in order to obtain a 7-point dose–response curve. Briefly, 5 µL of each antibody dilution was added to 45 µL of DMEM medium containing the pseudovirus and incubated for 1 h at 37 °C. Then, 50 µL of the antibody/pseudovirus mixture was added to each well and the plates were incubated for 24 h at 37 °C. Each point was assayed in triplicate. After 24 h of incubation, cell transduction was measured by the luciferase assay using the Bright-Glo™ Luciferase Assay System (Promega) and an Infinite F200 plate reader (Tecan). The measured relative light units (RLUs) were normalized with respect to the controls and the dose–response curves were generated and the neutralization dose 50 (ND50) was calculated with GraphPad Prism using a non-linear regression curve fitting. 

### 2.10. ELISA Binding and Competition Assay

The ELISA technique was used to evaluate the antibodies’ ability to bind RBD and compete with ACE2. The plates were coated with 50 µL per well of the RBD-6xHis protein, diluted in PBS at a final concentration of 1.42 µg/mL. The coating phase was carried out overnight at a temperature of 4 °C. Subsequently, the plates were washed three times with PBST (0.05% Tween 20/PBS) and then blocked with 100 µL of block solution (BSA 3% in PBST) for one hour at room temperature. This was followed by the competition phase, in which the supernatant of the hybridomas and ACE2-hFc at a concentration of 2 µg/mL in 1% BSA were incubated overnight at 4 °C. The plates were washed three times with PBST and, subsequently, incubated with the anti-human AP secondary antibody (Jackson 209-055-098) at a final dilution of 1:5000 in BSA, 1% PBST for 1 h. The plates were incubated with the pNPP alkaline phosphatase substrate for 1 h at room temperature, followed by reading at a wavelength of 405 nm. For the ELISA binding assay, plates were coated with 50 µL for the selected target, diluted to 1 µg/mL. The coating phase was carried out overnight at a temperature of 4 °C. Subsequently, the plates were washed three times with PBST and then blocked with 100 µL of block solution (BSA 3% in PBST) for one hour at room temperature. This was followed by the primary antibody incubation, in which the supernatant of the hybridomas or the purified mAb was diluted in 1% BSA–PBST and then incubated overnight at 4 °C. The plates were washed three times with PBST and, subsequently, incubated with anti-mouse AP (Southern Biotech 1030-04) or the anti-human AP secondary antibody (Jackson 209-055-098) at a dilution of 1:4000 in BSA, 1% PBST for 1 h. The plates were incubated with the pNPP alkaline phosphatase substrate for 1 h at room temperature, followed by reading at a wavelength of 405 nm.

### 2.11. Competition Assay Using Flow Cytometry

The ability of the antibodies to compete with the binding of RBD-hFc to the ACE2 protein expressed on Vero E6 cells (VERO C1008C1008 (clone E6, Vero E6) (ATCC^®^ CRL-1586™) was evaluated using flow cytometry. Briefly, the mAbs were diluted 1:2, 1:20, 1:200 in RBD-hFc (0.45 µg/mL) in FC buffer (PBS, 2 mM EDTA, 0.5% BSA) and incubated for 1 h at 4 °C. The 200,000 cells/well (96-well plate) were then stained with RBD-hFc or pre-incubated serum/RBD-hFc for 25 min at 4 °C. The cells were then washed with FC buffer and stained with the Alexa Fluor 488 conjugated anti-human IgG secondary antibody for 25 min at 4 °C. Finally, the cells were washed in buffer FC and resuspended in buffer FC before being run on the CytoFLEX flow cytometer (Beckman Coulter, Brea, CA, USA). Analysis was performed using the CytExpert 2.4 software (Beckman Coulter).

### 2.12. Size-Exclusion Chromatography

Size-exclusion chromatography (SEC) was performed using a Shimadzu Prominence system equipped with a quaternary solvent delivery pump, an SIL-20ACHT UFLC autosampler, a column heater set to 25 °C and an SPD-20AV ultraviolet (UV) detector set at 280 nm. The MAbPac™ SEC-1 size exclusion analytical column (5 µm, 300 Å, 7.8 × 300 mm) employed was provided by Thermo Scientific (Sunnyvale, CA, USA), whereas the TSK gel Super SW mAb HR analytical column (4 µm, 7.8 × 300 mm) was purchased from Tosoh Bioscience. For the separation of the antibodies, the MAbPac™ SEC-1 column was operated at a flow rate of 0.85 mL/min, and at 25 °C. The optimized SEC mobile phase was prepared with 50 mM sodium phosphate buffer at pH 6.8, containing 300 mM NaCl. The injection volume was 20 μL and the sample concentration was 1 mg/mL. For the separation of the RBD protein, the TSK gel Super SW mAb HR column was operated at a flow rate of 0.75 mL/min. The column heater was set to 25 °C, and the optimized SEC mobile phase was prepared with 50 mM sodium phosphate buffer at pH 6.8, containing 300 mM NaCl + acetonitrile (95:5, *v*:*v* respectively). The injection volume was equal to 20 μL. All buffers were filtered through 0.2 µm filters prior to use. Liquid chromatography instrument control and data analysis were performed using the Shimadzu LabSolutions 5.92 software. To estimate the sample purity, the data were analyzed in regard to the peak integration to calculate the peak area and the percentage of the peak area (Area%).

### 2.13. Biolayer Interferometry for Kinetics Studies

Binding studies were carried out using the Octet Red system (ForteBio, Las Vegas, NV, USA). All steps were performed at 25 °C, with shaking at 600 rpm in a 96-well plate (microplate 96 well, F-bottom, black, from Greiner Bio-One, Kremsmünster, Austria) containing 200 µL of solution in each well. A kinetics buffer 1X (ForteBio) was used throughout this study for the antibodies and analyte dilution and for washing the sensors. Kinetics assays were performed by first capturing the mAb using anti-mouse or anti-human Fc Octet biosensors (Anti-mouse IgG Fc Capture Biosensors, ForteBio, Fremont, CA, USA). The biosensors were soaked for 10 min in 1X kinetics buffer, followed by baseline signal measurement for 60 s and then loaded with murine mAbs (10 µg/mL) for 300 s (until the biosensor was fully saturated). After a wash step in 1X kinetics buffer for 120 s, the mAb-captured biosensor tips were then submerged for 300 s in wells containing different concentrations of the antigen (RBD-6xHis) to evaluate the association curves, followed by 900 s of dissociation time in the kinetics buffer. The binding curve data were collected and then analyzed using data analysis software v11.1 (ForteBio). Binding sensorgrams were first aligned at the beginning of the antigen binding cycle and following the single reference subtraction. The K_D_ values were calculated using a 1:1 global Langmuir binding model. The mAb captured biosensor tips were also dipped in wells containing the kinetics buffer to allow single reference subtraction to compensate for the natural dissociations of the captured mAb. 

### 2.14. Chimerization and Humanization 

Chimeric antibodies were generated by cloning the identified murine variable regions into expression vectors, encoding the human IgG1 heavy chain constant and the human light chain kappa constant domains. For humanization, the best matching (sequence identity) human germline sequences were identified using the IgBLAST program [24]. Structural templates to examine the original murine variable domains or the identified human germline sequences were identified through sequence searches against the PDB sequence database, either using the isolated light or heavy chain variable domain sequence or by searching for antibody structures with the best match for both the heavy and light chain variable domains. These structural templates were then superimposed using MICAN [25]. Non-conserved positions in the alignment between the original murine variable domain and the human germline sequences were then examined manually in the context of the PDB structural templates using the Discovery Studio Visualizer (BIOVIA, San Diego, CA, USA). Whenever a non-conserved position in the CDR regions appeared to be relevant in recognizing the antigen, for conformation of the CDR or the correct orientation of the interface between the heavy and light chain variable domains, the germline residue was replaced by the corresponding murine residue. 

### 2.15. Statistical Analysis

All data were statistically analyzed using Prism v9 (San Diego, CA, USA) and are represented as the mean ± SD. The IC50 values (50% inhibitory concentration) for the mAbs were calculated using non-linear regression, i.e., log(inhibitor) vs. normalized response variable slope.

## 3. Results

### 3.1. DNA and Recombinant Protein Immunization Induce High Titer of RBD-Specific Antibodies

DNA vaccination enables the electroporation and production of the desired antigen directly by the rodent muscle, exposing it to the immune system in its most natural conformation [22]. Therefore, in order to obtain functionally neutralizing anti-RBD antibodies, a vaccination approach based on DNA electroporation in skeletal muscles combined with protein injection was adopted. A schematic representation of the immunization schedule is shown in Figure 1A. According to the protocol, Balb/c mice were injected alternatively with a plasmid vector encoding a codon-optimized full-length spike protein (Figure 1B), and the recombinant RBD-hFc protein, both based on the Wuhan sequence. For the latter immunization modality, we designed a plasmid vector encoding RBD-hFc, then expressed and isolated it from mammalian Expi293 cells in a phosphate buffer suitable for immunization (Figure 1C). This combined immunization approach was used to produce antibodies that recognize the SARS-CoV-2 spike, particularly RBD, and interfere with binding to human ACE2. 

Finally, to identify the animals with the highest antibody titer to be used for the generation of anti-RBD antibodies, we performed an ELISA on day 27 after the first injection (Figure 1D). By combining the plasmid DNA-based genetic vaccination and the recombinant protein vaccination, all the 14 immunized mice showed an immunization titer > 1:291,600. 

### 3.2. Identification of Potent High-Affinity RBD-Specific Antibodies from Hybridomas

Two weeks after the last immunization, all the mice were euthanized, and the spleens and lymph nodes processed for fusion with myeloma cells. The resulting hybridomas were screened using ELISA to identify RBD-specific mAbs and then analyzed by HT-SPR to identify the one with the higher affinity.

The hybridomas obtained from splenocyte/myeloma fusion were screened using ELISA and more than 2000 wells were found to be positive for WT RBD-specific mAbs. Subsequently, an SPR-based kinetic profile screening of the hybridoma cell culture supernatant was performed using the LSA high-throughput SPR (HT-SPR) system. The anti-RBD mAb samples were diluted and captured onto individual spots, where the chip was activated using a goat anti-mouse IgG Fc. Following the kinetic characterization of the antibodies, a selection of high-affinity antibodies were identified. 

An iso-affinity plot (Figure 2A) depicts the relationship between the association rate constant (k_a_, y-axis) and the dissociation rate constant (k_d_, x-axis) of all the mAbs measurements, in which every dot represents the measurement for a single hybridoma. All the hybridomas with an affinity < 380 pM, displaying a k_d_ < 1 × 10^−4^ s^−1^ and a k_a_ > 5 × 10^4^ M^−1^s^−1^ were selected for further analysis (Figure 2A). The values of these parameters have been defined according to the affinity of the ACE2/RBD interaction (44 nM [26]), showing that the selected hybridomas display an affinity more than 100-fold higher than the ACE2/RBD interaction, with a high percentage of competing antibodies. To assess the productivity of each selected hybridoma, the mAb present in the culture medium was quantified by biolayer interferometry (BLI) using the Octet RED96 system (ForteBio), using anti-mouse Fc biosensors. The results show that the concentration of IgG in the cell culture media supernatant was between 1 and 50 µg/mL (Figure 2B). Using the HT-SPR system, it has been possible to select those hybridomas that express high-affinity RBD-specific antibodies and a set of 427 antibodies with affinity values between 10 and 100 pM was identified. The outcome of these results involved the successful selection of high-affinity RBD-specific antibodies from hybridomas, with affinities over 100-fold greater than the ACE2/RBD interaction, and productive antibody concentrations in the culture medium ranging from 1 to 50 µg/mL.

### 3.3. Potent Neutralization of SARS-CoV-2 by High-Affinity Antibodies in VSV Pseudovirus Assay

The high-affinity antibodies previously selected were then analyzed for the ability to neutralize the SARS-CoV-2 spike (Wuhan strain) in a VSV-based pseudovirus assay. For the neutralization assay, Vero E6 cells were plated in 96-well plates and, after 24 h, the hybridoma supernatants were diluted in PBS, mixed with Fluc expressing the pseudovirus (1:1, *v*/*v*), incubated at 37 °C for 30 min and added to the cells in duplicates. After 16–18 h of incubation, the transduction efficiency was analyzed.

The hybridomas were initially screened at three fixed concentrations (Figure 2C) and then the most active (about 70 mAbs) were tested using a 10-point dilution curve. The obtained IC50 values ranged between 0.01 µg/mL to a maximum value of 0.25 µg/mL (Figure 2D). These results demonstrate the potent neutralizing capability of the selected high-affinity antibodies against the SARS-CoV-2 spike, and a set of about 50 mAbs with higher IC50 values have been identified for further characterization.

### 3.4. Highly Competitive Monoclonal Antibodies against SARS-CoV-2 Spike Interaction with hACE2 Revealed

To develop mAbs that effectively prevent SARS-CoV-2 infection, as well as infection by other coronaviruses through human angiotensin-converting enzyme 2 (hACE2), it is also of great importance to gain a comprehensive understanding of how the virus interacts with its target, hACE2. To have a complete overview of the binding and competing properties of the previously selected highly neutralizing mAbs, competition assays using both flow cytometry and ELISA were also carried out. First, we produced and purified the hACE2-hFc fusion protein and WT RBD-6xHis in Expi293 cells, and verified their correct folding state and the absence of aggregates with both Coomassie staining and SEC (Figure 3A,B). To validate the binding properties of the most active mAbs and provide complementary information to previous SPR screening data, the recombinant proteins obtained were then used to perform a competition ELISA, as described in Figure 3C. The plates were coated with the RBD-6xHis protein and then incubated with a serially diluted supernatant of the hybridomas in the presence of the hACE2-hFc protein. Through this assay, we were able to determine the IC50 values (Figure 3D), and most of the antibodies showed an IC50 value between 0.1 and 0.8 µg/mL. To support this data, Vero cells expressing hACE2 were used to perform a FACS competition assay, incubating them with RBD-hFc alone or RBD-hFc pre-incubated for 1 h with the selected mAbs at three serial dilutions (Figure 3E). The cells were then stained with the Alexa Fluor 488 conjugated anti-human IgG secondary antibody and analyzed using FACS to detect the binding of RBD-hFc to hACE2. As expected, most of the antibodies were found to compete with the binding between RBD and hACE2 (Figure 3E).

The results showed that the selected mAbs, characterized by their competitive binding to the SARS-CoV-2 spike protein and their ability to impede its interaction with hACE2, exhibit strong potential for preventing not only SARS-CoV-2 infection, but also infections by other coronaviruses utilizing the ACE2 receptor. This comprehensive analysis, combining flow cytometric assays, competition ELISA and IC50 determination, underscores the antibodies’ robust binding properties and their potential as effective countermeasures against viral infections mediated through ACE2 interactions.

### 3.5. Selected Monoclonal Antibodies Are Versatile and Bind with High Affinity to SARS-CoV-2 Spike and Variants

A further selection based on affinity, neutralization power and competing activity led to a set of about 20 antibodies. The selected mAbs were then evaluated for their binding to the SARS-CoV-2 spike in its natural conformation and the ability to interact with multiple RBD variants. To assess the binding of these antibodies to the full-length spike protein, HEK-293 cells were transfected with an empty vector or SARS-CoV-2 spike and treated with antibodies. As shown in Figure 4A, the binding percentage of a selected panel of murine mAbs is comparable or even higher than the commercially available anti-spike antibody used as the control, confirming the affinity of our mAbs to the spike.

In parallel, to evaluate the ability to recognize multiple variants, kinetics assays were performed on a panel of variants, which is critical to understanding the functional efficiency of SARS-CoV-2 antibodies. The K_D_ of the selected antibodies on the variants of interest were subsequently evaluated by BLI analysis. The results indicate (Figure 4B) that most of the antibodies tested bind to all the variants with a K_D_ between 10^−10^ M and 10^−8^ M. Regrettably, some antibodies showed reduced affinity for the S477N variant. However, the following antibodies, namely 9-8F2-B11, 2-12C7-B3, 9-2H7-D7, 9-8H7-B4 and 9-3H9-3E9-B12, were capable of binding to the S477N mutant. The same set of antibodies were tested by ELISA to evaluate their EC50 values (Table 1 and Figure 4C). The plates were coated with RBD-6xHis WT (D614), N501Y (B 1.1.7), N439K, S477N, N501Y-E484K-K417N (B.1.1.351), L452R, N501Y-E484K-K417T (P.1) or WT FL spike and then incubated with serially diluted, purified antibodies. The results pertaining to the WT FL spike are summarized in Table 1 and reveal that some of the antibodies exhibited remarkably low EC50 values, with the 2-12D6-G6 antibody registering a value as low as 0.1 ng/mL. The majority of antibodies showed an EC50 value between 2.5 and 6 ng/mL. The results for RBD WT and the variants are summarized as a heat map (Figure 4C) and show that selected antibodies bind most of the variants, with EC50 values between 10 and 20 ng/mL, but with a reduced binding capacity for the S477N variant, confirming the data obtained from the BLI analysis. 

Among these antibodies, 9-8F2-B11 emerged as a standout, exhibiting remarkable binding to selected variants. To appropriately evaluate the binding in a conformational-type assay and its ability to recognize multiple variants, the selected antibody was also analyzed using flow cytometry. Flow cytometry was performed after transfecting HEK-293 cells with a plasmid encoding the FL spike of D614G, Alpha and Beta variants. The outcomes of these experiments demonstrated that 9-8F2-B11 exhibited high efficacy in binding to all three variants (Figure 4D).

The results showed that the selected mAbs demonstrated remarkable binding affinity not only to the full-length SARS-CoV-2 spike protein, but also to multiple viral variants, including D614G, Alpha, and Beta variants, as revealed by flow cytometry and ELISA. These findings emphasize the antibodies’ broad recognition and high affinity to various spike protein variants, highlighting their potential as versatile therapeutic candidates against SARS-CoV-2. 

### 3.6. Selection and Humanization of Potent SARS-CoV-2 Neutralizing Antibody

Our approach led to the selection of several antibodies whose binding properties potentially have the ability to neutralize SARS-CoV-2, both the Wuhan and several variant strains. Among the antibodies characterized, 9-8F2-B11 showed the most interesting characteristics in terms of binding and neutralizing activity, so this was subjected to chimerization and humanization to make it suitable for clinical purposes and was further characterized. Starting from the original variable domain sequence of the murine hybridoma antibodies, chimeric and humanized antibodies were synthesized and produced using high-throughput transient transfection of high-density Expi CHO cells. The murine variable regions of 9-8F2-B11 were cloned in frame, with constant human IgG/kappa constant regions, to obtain the corresponding chimera. Once produced by transient transfection, the purity of the antibody was evaluated using the SDS-PAGE method and western blot analysis (Figure 5A), conducted in reducing conditions, and the ability to bind recombinant RBD was assessed using ELISA (Figure 5B). The chimeric antibody was then subjected to humanization and several variants were designed with different levels of humanization (Appendix A). Through western blot analysis (Figure 5C) and ELISA (Figure 5D) it was found that some combinations of the humanized 9-8F2-B11 VH and VL domains, although having very similar functional characteristics, were more efficiently produced than others, an essential requirement for their large-scale production for clinical use. For this reason, the humanized VH_3_VL_3_ combination was selected as the final lead. F_c_ and F_v_ modification processes sometimes trigger significant changes in the binding and affinity of the final chimeric and humanized antibody [27,28]. Indeed, kinetic analysis (Figure 5E) showed that the humanized VH_3_VL_3_ combination of 9-8F2-B11 significantly loses affinity for the B.1.1.351 and BA.1 variant compared to the murine version. After performing all the analyses, the results showed that the murine 9-8F2-B11 was successfully humanized and the VH3VL3 combination was chosen as the lead, despite some affinity changes for specific variants during the humanization process.

### 3.7. Successful Selection of a Second Antibody That Can Neutralize the Omicron BA.1 Variant

Effective antibody binding and affinity are not necessarily a guarantee of neutralizing activity against viral particles, and pseudovirus neutralization assays are essential for full characterization. After confirming the reduced neutralizing activity of 9-8F2-B11 against the Omicron BA.1 variant due to lower affinity, we initiated a new screening campaign using pseudovirus neutralization assays with the complete hybridoma supernatant panel. Capitalizing on our extensive antibody library, we aimed to expediently identify antibodies capable of neutralizing the Omicron BA.1 variant. The supernatants were tested using a single dilution and single replicate to identify neutralizing and effective antibodies against BA.1. The results obtained indicated a group of about 30 hybridomas that reduced the spike-mediated entry rates of viral particles into cells by up to 60% or more, also demonstrating promising neutralizing potency against the BA.1 variant (Figure 6A).

Among these, we selected 3-12B12-F4, an antibody that we further confirmed efficiently binds and neutralizes pseudoparticles of SARS-CoV-2 BA.1. This antibody was purified from the supernatant and then analyzed again using a pseudovirus assay. Together with 9-8F2-B11, 3-12B12-F4 was then further analyzed in decreasing dilutions on several SARS-CoV-2 variants (Figure 6B), including the BA.1/BA.2/BA.5 variants. Interestingly, while 9-8F2-B11 maintained its neutralizing ability in the nanomolar range for all tested variants except for BA.1, 3-12B12-F4 efficiently neutralized the latter. In addition, unexpectedly, 9-8F2-B11 instead neutralized the Omicron BA.5 variant, while 3-12B12-F4 showed reduced activity against both BA.5 and Delta (B1.617.2). Thus, this second screening identified a second antibody (3-12B12-F4) that had different, but complementary, neutralizing abilities to 9-8F2-B11. To assess their combined neutralizing breadth, we tested combinations of the two antibodies, both in their murine (Figure 6C) and chimeric formats (Figure 6D). Thus, for both formats, when combined, they have the ability to neutralize all the variants tested within the nanomolar range. The results underscore the significance of pseudovirus neutralization assays in evaluating antibody neutralization, and the diverse antibody library previously generated facilitated the rapid identification of 3-12B12-F4 as a potent neutralizer of Omicron BA.1, enhancing the neutralizing capacity alongside 9-8F2-B11.

### 3.8. Antibodies 9-8F2-B11 and 3-12B12-F4 Bind to Different Sites

In order to identify the binding region of selected mAbs on RBD, docking analysis was adopted as a preliminary investigation strategy. Models of mAb and RBD were used to sample the binding conformations via three different docking algorithms (HADDOCK [29,30], ZDOCK [31], FRODOCK [32]). The solutions obtained were pooled and ranked using the Consrank algorithm [33], which selected the most probable binding conformation based on the conservation of the interacting residues across the various solutions. This led to the identification of nine possible contact residues (L452, E484, F486, Q493, S376, D405, R408, G446, G496), which were mutated in Omicron variants compared to the WT RBD. To test this hypothesis, we produced and purified the corresponding single mutant proteins from Expi293 cells (Figure 7A), and an ELISA was performed to assess which single mutation was responsible for the loss of binding of the murine 3-12B12-F4 and 9-8F2-B11 antibodies (Figure 7B,C). The binding of both antibodies was not affected by the single mutations, namely E484A, F486V, Q493R, D450N, R408S, G496S, so, taken individually, these mutations do not determine recognition. For the L452R and G446S mutations, on the other hand, there was a significant decrease in binding for 3-12B12-F4 and 9-8F2-B11, respectively, clearly indicating their involvement in the antibody binding site.

Neutralizing antibodies (NAbs) that target the RBD of SARS-CoV-2 have been categorized into four structural groups, referred to as C1 to C4 [34]. This categorization is based on their ability to bind to the RBD in either its up or down conformation and their proximity to the ACE2 binding site. The C1 group encompasses NAbs that insert themselves into the RBD’s ACE2 binding site, effectively mimicking the ACE2 binding process. As a result, these antibodies exclusively bind to the RBD when it assumes the up conformation. NAbs belonging to the C2 class obstruct the ACE2 binding site, while retaining the capability to bind to the RBD in both its up and down conformations. Conversely, C3 NAbs primarily attach to regions of the RBD situated outside the ACE2 binding site. This binding occurs regardless of whether the RBD is in the up or down conformation.

To potentially categorize the 9-8F2-B11 and 3-12B12-F4 antibodies and uncover potential epitopes, we employed a computational approach involving the analysis of their variable domains (Fv) docked against residues known to impair binding. This analysis was conducted using the PDBePISA tool, which provided valuable insights into the interactions between these antibodies and the receptor-binding domain (RBD) of the SARS-CoV-2 spike protein. The detailed interaction network that emerged from this analysis is summarized in Table 2.

Our analysis revealed that both the heavy chain (VH) and light chain (VL) variable domains of 9-8F2-B11 and 3-12B12-F4 played integral roles in binding to the RBD, forming multiple hydrogen bonds. Notably, 9-8F2-B11 also established additional salt bridges.

Further examination of the interactions between 3-12B12-F4 and RBD showed that this antibody bound to residues situated on the opposite side of the ACE2 receptor-binding motif (RBM). These residues included G482, N450, I472, Y351, S349, G447 and E480 (as detailed in Table 2). This unique binding pattern aligns with the classification of 3-12B12-F4 as a C3 antibody.

In contrast, the interactions of 9-8F2-B11 involved both its VH and VL with the RBD, forming hydrogen bonds and salt bridges. Remarkably, these interactions extended to the binding of K444 within the RBD. The model generated based on these in vitro binding experiments, in conjunction with PDBePISA analysis, indicates that 9-8F2-B11 covers a substantial portion of the RBM, albeit not at the central epitope. Based on these observations and structural insights obtained through the PDBePISA, we classify 9-8F2-B11 as a C1 antibody.

In conclusion, our comprehensive structural studies, epitope mapping and NAb classification unveil the unique binding properties of 3-12B12-F4 as a C3 NAb and 9-8F2-B11 as a C1 NAb, shedding light on their potential use in the fight against SARS-CoV-2 and its variants.

## 4. Discussion

Monoclonal antibodies have been recognized as crucial tools in antiviral therapy and are now quickly making their mark in supplementing vaccines in the fight against newly emerging pathogenic viruses, such as SARS-CoV-2. To develop mAbs that effectively neutralize SARS-CoV-2 infection, it is of fundamental importance to understand how the virus interacts with its cellular receptor, the angiotensin converting enzyme 2 (hACE2), a crucial protein for the entry of the virus into the cell [35]. Viral entry depends on the spike glycoprotein, one of the structural proteins that decorate the surface of SARS-CoV-2. This protein is divided into two different subunits, S1 and S2, as follows: the first is responsible for binding to ACE2 through the RBD, while the second, the S2 subunit, mediates the fusion of the viral cell membrane to that of the host cell. Due to its clear role of “initializing” viral infection, the spike protein quickly became the main molecular target to be neutralized with antibodies and the focus of therapeutic tools [36]. In particular, the RBD region has been widely utilized as the target of choice in developing many of the mAbs against SARS-CoV-2.

In this study, we immunized BALB/c mice with both pDNA expressing the SARS-CoV-2 spike protein and the recombinant RBD protein. Electro-gene-transfer in mice skeletal muscles by itself is able to elicit humoral and cellular immune responses, having adjuvant-like properties in vaccination when coupled with plasmid DNA injection [37,38]. The combination of genetic immunization and intraperitoneal protein vaccination allowed us to generate a diverse panel of nearly 2000 antibodies specifically targeting the RBD protein of SARS-CoV-2. In this study, we identified two extremely promising mAbs against the SARS-CoV-2 RBD protein. Given their exceptionally high affinity for the SARS-CoV-2 RBD, these antibodies have the potential to be harnessed for virus neutralization and developed for therapeutic applications. Towards the end of 2020, following nearly a year of circulation among humans, the SARS-CoV-2 underwent a significant transformation in its ability to thrive within the human population. These extensively mutated versions of SARS-CoV-2 displayed increased transmission rates compared to earlier variants and earned the classification of ‘variants of concern’ (VOCs). The triumph of each VOC over the preceding dominant variant can be ascribed to alterations in the virus’s intrinsic functional traits. Furthermore, these VOCs demonstrated varying degrees of modifications in virus antigenicity, equipping them with the capability to either evade a pre-existing primed immune response or resist antibody therapies. The current circulating variant is represented by the BA.2.75 and XBB.1.5 lineage, which are classified as ‘variants of interest’ (VOIs). Indeed, emerging SARS-CoV-2 variants evading currently available antibody and vaccine therapies are of considerable concern. SARS-CoV-2 Omicron sublineages BA.4 and BA.5 are the variants previously circulating worldwide with the classification of VOCs. Here, the lack of the G496S mutation in their spike protein increased their hACE2 binding affinities compared to other Omicron sub-variants [39,40]. Several potent neutralizing antibodies are in clinical use or in clinical trials. However, these variants exhibit increased resistance to these antibodies. Bamlanivimab (LY-CoV555), etesevimab (LY-CoV016) and casirivimab (REGN10933) are reported to be sensitive to the mutations E484K and L452R; K417N/T; and E484K and K417N, respectively [41,42]. Furthermore, the B.1.351 variant, which carries K417N/E484K/N501Y mutations in S331-524, is refractory to neutralization by Bamlanivimab (LY-CoV555), etesevimab (LY-CoV016) and casirivimab (REGN10933) [43,44]. In this context, as new variants emerge, the combinatorial action of different antibodies may create an efficient cocktail to block viral infection, which also acts on rapidly evolving neutralization-escape mutants.

Our work presents a library from which we were able to select two mAbs, that used in combination, effectively neutralized all the tested variants. The availability of this antibody library opens up the possibility of rapid screenings and the identification of antibodies effective against newly emerging variants of the corresponding pathogen, thus significantly accelerating the drug discovery process. More importantly, some of these antibodies are also effective in in vivo pseudovirus assays, in K18 hACE2 transgenic mice, and are capable of cross-binding SARS-CoV-1, as we have observed in an additional preliminary screening (Appendix A). These additional results highlight the enormous potential of this antibody panel for cross-neutralization of other Sarbecoronaviridae members.

This approach offers a promising foundation for addressing current and future coronavirus pandemics, as it is crucial to prepare for the potential emergence of new coronaviruses from zoonotic sources.

## 5. Conclusions

In summary, our study focused on identifying potential neutralizing antibodies against SARS-CoV-2 and its variants, resulting in the discovery of two antibodies with extremely high affinity, binding capacity and inhibitory efficacy against the virus in vitro in its wild-type and mutated forms. These antibodies, with their remarkable attributes, have great potential to be developed as therapeutic tools. Additionally, our extensive panel of mAbs was shown to be a valuable resource for screening, identifying and developing new neutralizing Sarbecoronavirus antibodies.

## 6. Patents

IT202100023816A1

## Figures and Tables

**Figure 1 antibodies-13-00005-f001:**
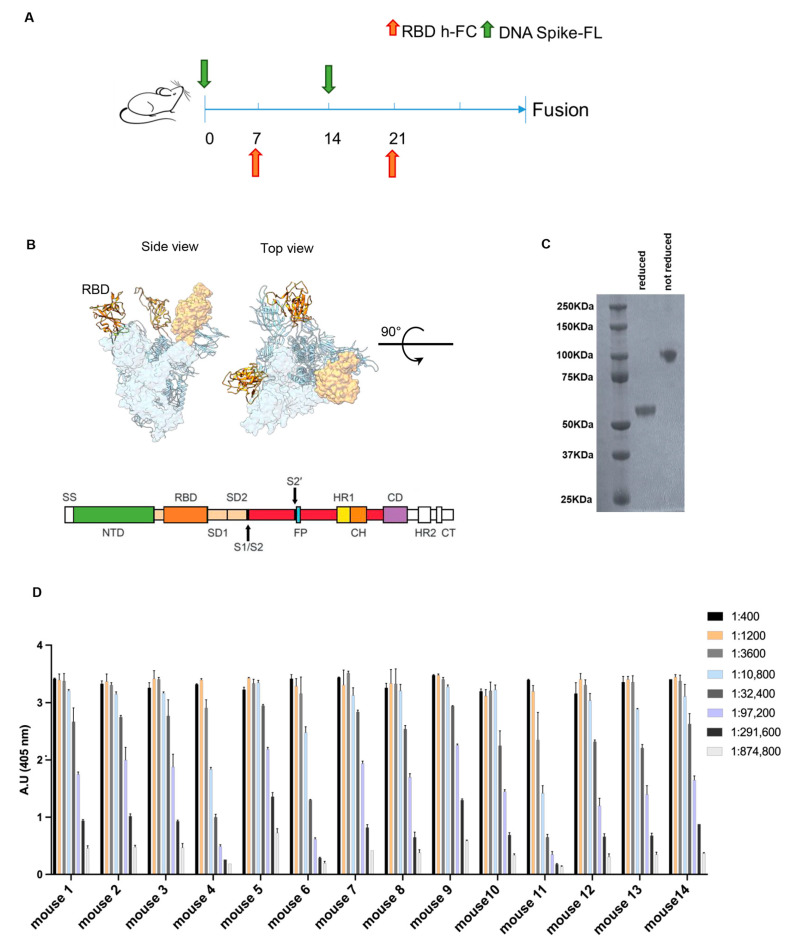
Combined genetic and protein immunization approach for antibody generation to RBD. (**A**) Time schedule of the immunization plan. Green arrow represents injection of DNA spike FL at day 0 and 14 the orange one represents the RBD-hFc injection at day 7 and 21. (**B**) In the upper section, an overview of the spike FL protein from the side and top view is provided (RBD in orange); in the lower section, a schematic representation of the main linear domains encoded by the DNA spike FL used for the immunization plan is provided. (**C**) The Coomassie stain acquisition of the RBD- hFc protein in reduced and non-reduced form is provided. The expected MW 55 KDa (**D**) ELISA against the RBD-6xHis protein to evaluate the immunized mice immune titer. Values reported are serial dilutions of sera; the results are shown at 405 nm absorbance.

**Figure 2 antibodies-13-00005-f002:**
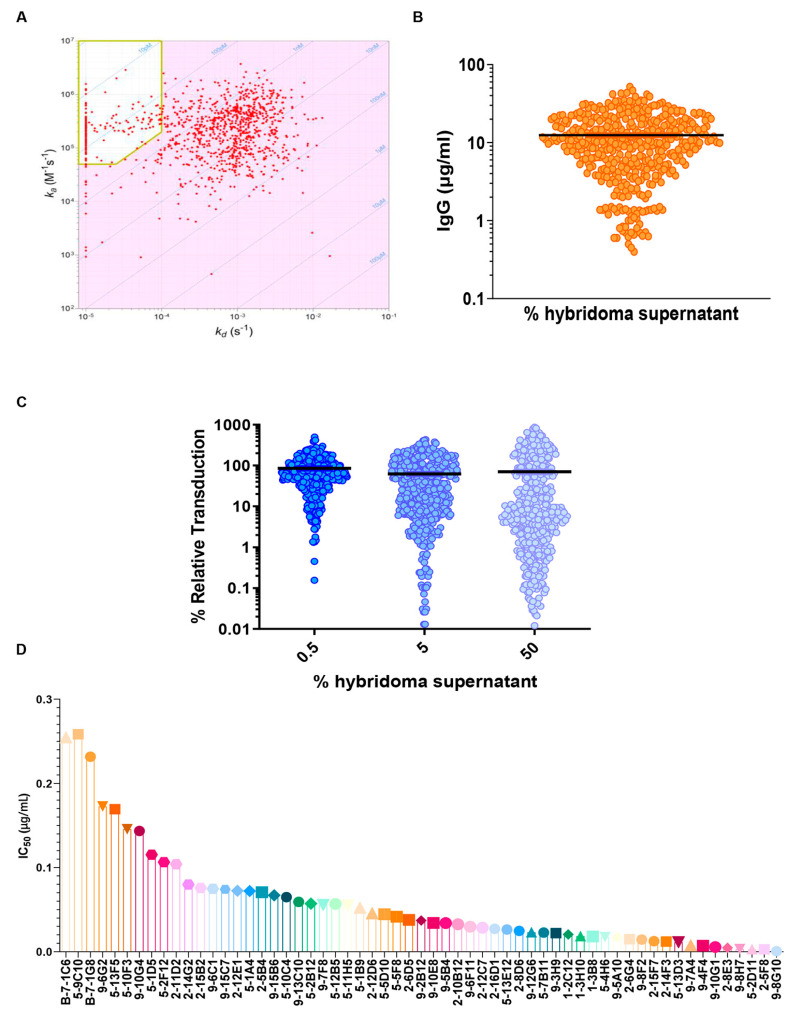
Massive characterization of huge panel of hybridoma supernatants. (**A**) Iso-affinity plot of hybridomas at a single dilution to evaluate the ka and kd. In the yellow box, the selected antibodies have better binding (affinity < 380 pM, displaying a kd <1 × 10^−4^s^−1^ and a ka >5 × 10^4^ M^−1^s^−1^). (**B**) Total IgG amount for each analyzed sample (mg/mL) using Octet RED96 system (ForteBio). (**C**) VSV(Fluc_eGFP)-CoV2-PP pseudovirus assay using SARS-CoV-2 Wuhan variant. All hybridoma supernatants were screened in duplicates, with 3 concentration steps to evaluate their neutralization properties. (**D**) Obtained IC50 values for pseudovirus assay; the obtained values swing between 0.01 µg/mL to a maximum value of 0.25 µg/mL.

**Figure 3 antibodies-13-00005-f003:**
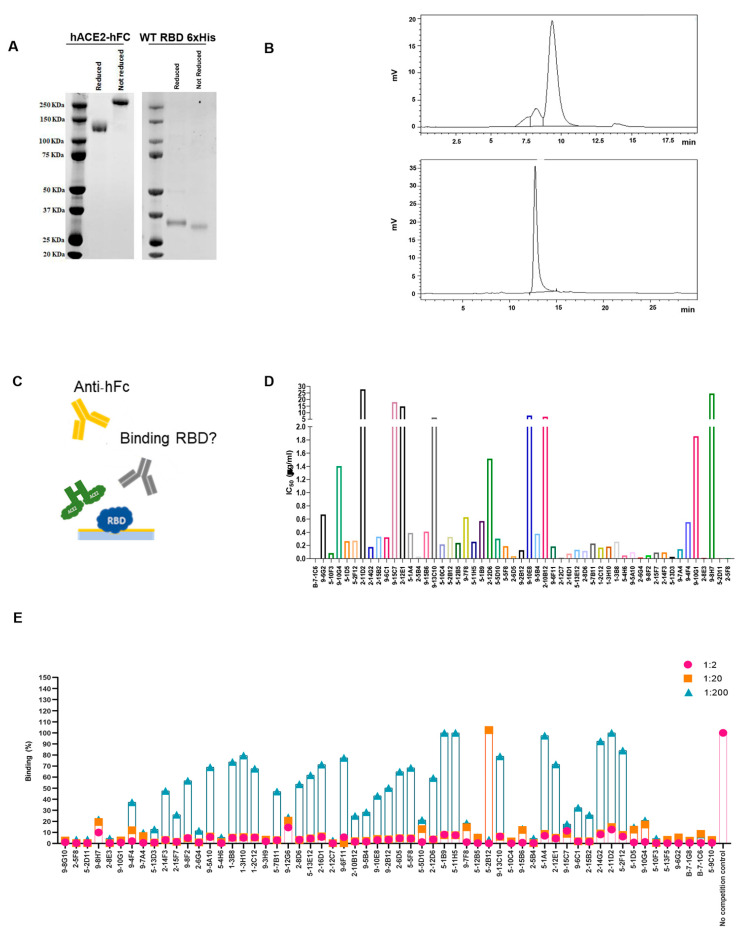
ELISA and FACS assays to characterize hybridoma supernatant properties. (**A**) Coomassie staining of purified hACE2 protein and WT RBD-6xHis in reducing and non-reducing conditions. Both produced proteins have been used in subsequent ELISAs. MW RBD-6xHis 26 KDa, MW hACE2 145 KDa. (**B**) SEC analysis of the hACE2 (upper) and WT RBD-6xHis (lower) to evaluate purity of the produced proteins. (**C**) Schematic representation of the competition ELISA assay performed on selected hybridoma supernatants. (**D**) Competitive ELISA, results are shown as A.U. (405 nm). High competing mAbs display low IC50 values. (**E**) FACS assay on hACE2-expressing Vero cells, using hybridomas culture media in three serial dilutions (1:2, 1:20, 1:200). Antibodies neutralizing the binding display 0% binding percentage.

**Figure 4 antibodies-13-00005-f004:**
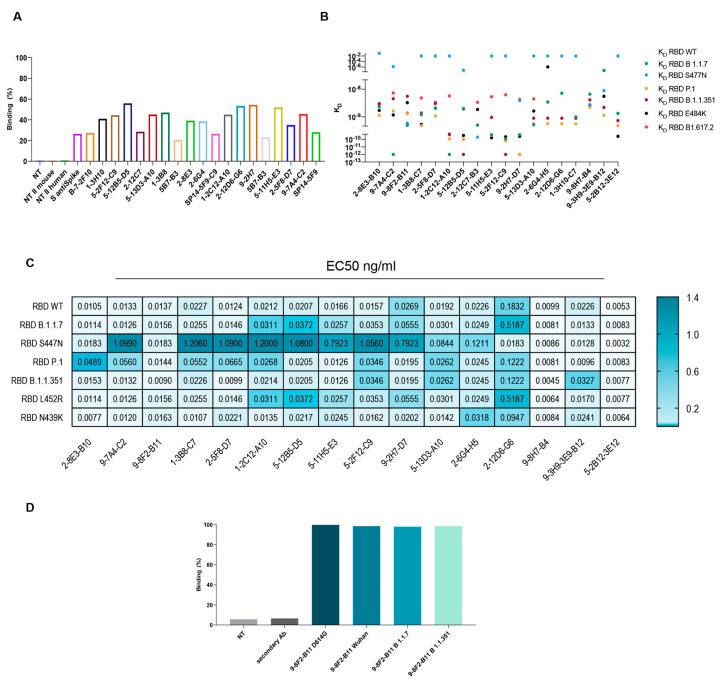
Characterization of mAb binding to SARS-CoV-2 spike and its variants. (**A**) FACS assay on HEK-293 cells transfected with spike FL for a selected group of purified antibodies. S anti-spike: commercial control antibody. (**B**) BLI assay to evaluate K_D_ of the selected mAbs on a specific set of VOC. (**C**) ELISA on purified mAbs using produced spike FL or RBD variants to evaluate EC50 (ng/mL). (**D**) FACS assay on HEK-293 cells transfected with spike FL variants to evaluate binding of purified 9-8F2-B11 mAb. Cells were treated with purified antibodies to a final concentration of 1 µg/mL.

**Figure 5 antibodies-13-00005-f005:**
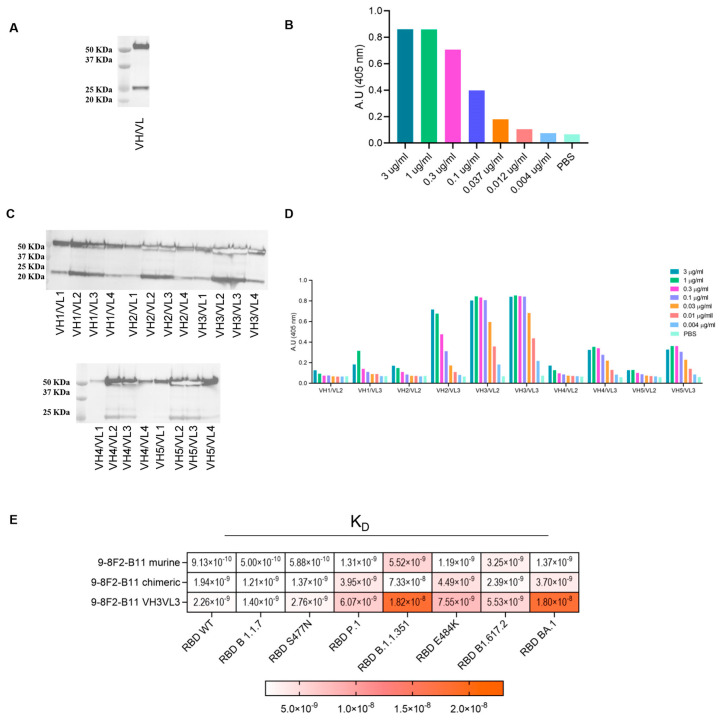
Assessment of binding and productivity of chimeric and humanized 9-8F2-B11 antibodies. (**A**) Western blot analysis of 9-8F2-B11 chimera staining VH (MW 50 KDa) and VL (MW 25 KDa) fragments. (**B**) ELISA using 9-8F2-B11 chimera on WT RBD protein. Results are shown as A.U. (405 nm). (**C**) Western blot analysis of high-throughput production of VH (MW 50 KDa) and VL (MW 25 KDa) combinations of humanized 9-8F2-B11 to evaluate their productivity. (**D**) ELISA testing conducted on RBD WT protein to assess the performance of the best-producing humanized mAbs. The combination featuring VH3VL3 was selected for humanized 9-8F2-B11. Results are shown as A.U. (405 nm). (**E**) BLI assay to evaluate K_D_ values of the selected 9-8F2-B11 murine, chimeric and humanized mAbs on a specific set of variants.

**Figure 6 antibodies-13-00005-f006:**
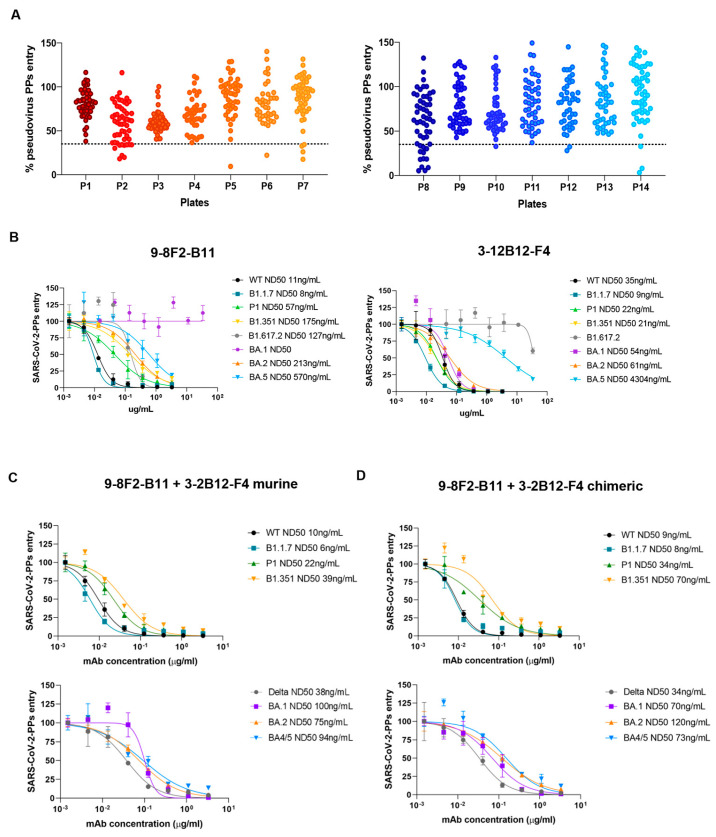
Screening for neutralizing antibodies against Omicron variant BA.1 and evaluation of their combinations on selected variants. (**A**) Antibody library single dilution screening on BA.1 Omicron variant. Supernatants were tested in a total of 14 96-well plates. Dotted line represents the chosen cut-off for the identification of neutralizing mAbs. (**B**) Murine 9-8F2-B11 and 3-12B12-F4 tested on pseudovirus assay for selected variants. ND50 values obtained for each variant are reported in the legend. (**C**) Pseudovirus assay using combinations of murine 9-8F2-B11 and 3-12B12-F4. (**D**) Pseudovirus assay using combinations of chimeric 9-8F2-B11 and 3-12B12-F4.

**Figure 7 antibodies-13-00005-f007:**
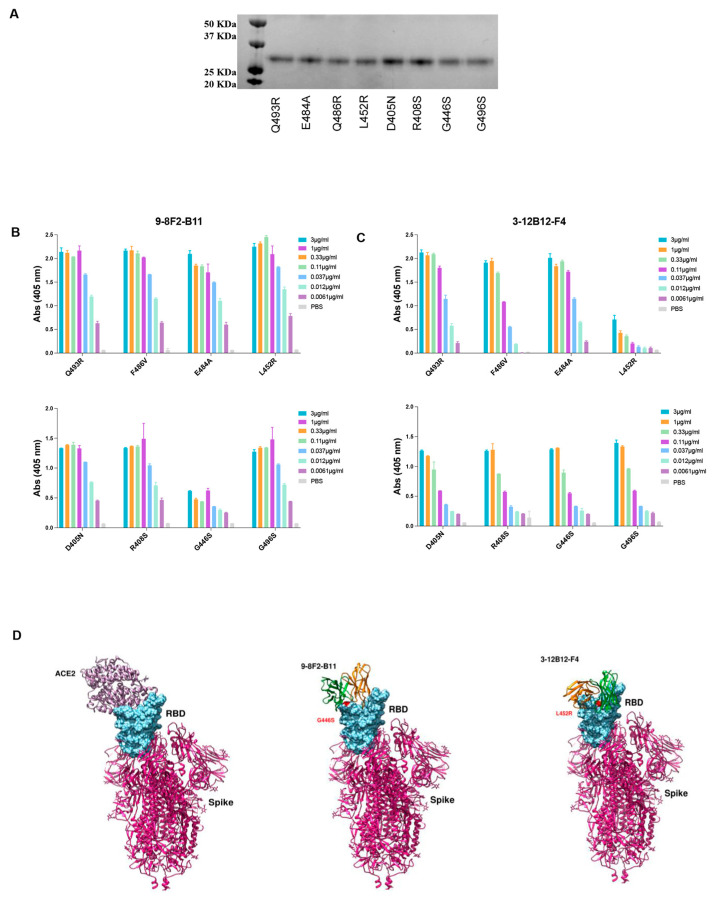
Binding analysis of antibodies 9-8F2-B11 and 3-12B12-F4 to RBD variants. (**A**) Coomassie staining of the RBD single mutant. Expected MW 26 KDa. (**B**) ELISA to evaluate binding of 9-8F2-B11 to the single mutants. Results are shown as A.U. (405 nm). (**C**) ELISA to evaluate binding of 3-12B12-F4 to the single mutants. Results are shown as A.U. (405 nm). (**D**) Three-dimensional structure simulation and docking analysis of ACE2, 9-8F2-B11 and 3-12B12-F4.

**Table 1 antibodies-13-00005-t001:** ELISA on spike FL EC50 values. Calculated EC50 (ng/mL) values for each tested antibody on the WT spike FL.

mAb	EC50 on Spike (ng/mL)	mAb	EC50 on Spike (ng/mL)
2-8E3-B10	2.93	5-11H5-E3	3.59
9-7A4-C2	2.65	5-2F12-C9	4.42
9-8F2-B11	3.68	9-2H7-D7	4.57
1-3B8-C7	2.62	5-13D3-A10	4.78
2-5F8-D7	2.55	2-6G4-H5	4.58
1-2C12-A10	3.39	2-12D6-G6	0.11
5-12B5-D5	3.53	2-12C7-B3	5.79

**Table 2 antibodies-13-00005-t002:** List of interaction residues between RBD and selected mAbs, extracted from docked models using PDBePISA. Interacting pairs are divided into h-bonds (blue) and salt bridges (yellow). In green and orange, the interacting residues of VH and VL chains of each mAb. Bond lengths are reported in the middle column of each pair. HE1/2 and OD1/2 stand for epsilon hydrogens of Gln and delta oxygens of Asp, respectively.

	RBD	Dist (Å)	9-8F2-B11
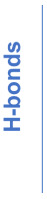	L452	2.05	N35
	Q493 (HE1)	2.24	K55
	Q493 (HE1)	2.44	K55
	Y449	2.34	R50
	K444	1.54	D57
	K444	1.56	D52
	G446	3.19	E31
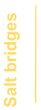	K444	3.72	D57 (OD1)
	K444	2.55	D57 (OD2)
	K444	2.60	D52
	**RBD**	**Dist (Å)**	**3-12B12-F4**
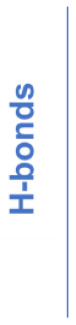	G482	3.03	S58
	N450	2.95	V106
	L452	3.27	N57
	Y351	1.79	S103
	S349	3.01	S103
	N450	2.01	D49
	G447	1.93	Y31
	C480	1.73	Y93

## Data Availability

The datasets used and/or analyzed during the current study are available from the corresponding author on reasonable request.

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
