# Peer review of "Isolation and Characterization of Neutralizing Monoclonal Antibodies from a Large Panel of Murine Antibodies against RBD of the SARS-CoV-2 Spike Protein"

_2073-4468, 2024, doi:10.3390/antib13010005_

Round 1

Reviewer 1 Report

Comments and Suggestions for Authors

The manuscript by Muzi et al is written in such a wonderful way. Although they still need to be validated in preclinical animal model studies involving nonhuman primates and randomized proof-of-concept clinical trials, the authors have succeeded in developing potentially life-saving monoclonal antibodies (mAbs), an arsenal to fight COVID-19. Additionally, they have applied for patents for their discoveries and received approval, IT202100023816A1.

Although I much like their accomplishments and the quality of their work, I feel that the following points should be addressed to improve the quality and readability of their manuscript further.

1.  How long did take the authors to develop their mAbs?  Why did authors choose to make their mAbs using conventional technology? Why did they not make use of faster technology, like recombinant antibody production? If the authors had taken a faster route, these mAbs might have prevented millions of COVID-19 deaths during the pandemic.

2.  The methodological section's subsections need to be restructured to achieve coherence in the experimental steps. For instance, all of the recombinant protein production, purification, and characterization experiments as well as the DNA plasmid construction experiments were completed before the experiments on immunization and hybridoma development.

3.   In vivo efficacy data from K18 hACE2 transgenic mice is lacking despite it being mentioned in the manuscript of lines 791-792. Why were the data not disclosed by the authors? If available, this data should be included in the manuscript as it is very important.

4.  Some aberrations are repeated multiple times in their expanded form. Shortened from can be utilized throughout the manuscript once they have been defined in the first use. This can save space; therefore, authors should pay attention to it.  The following are some examples I noticed in the manuscript.

a. Monoclonal Antibodies (mAbs) is shown in lines 49, 475, 509, 530, 737, 750, 757, 785, and 804.

b. The Receptor-Binding Domain (RBD) is shown on lines 710–745.

c. ACE2 Receptor-Binding Motif (RBM), lines 716, 723.

d. Line 743: the glycoprotein Spike (S) 

e. Line 761: severe acute respiratory syndrome coronavirus 2 (SARS-CoV-2)

Author Response

For Research Article “Isolation and characterization of neutralizing monoclonal antibodies from a large panel of murine antibodies against RBD of the SARS-CoV-2 Spike protein.”

Response to Reviewer 1 comments:

We would like to extend our sincere gratitude to the reviewer for their kind words and for acknowledging the importance of our research efforts. We truly appreciate the recognition of our work in developing potentially life-saving monoclonal antibodies to combat COVID-19. We also share the reviewer's excitement about the recent patent approval (IT202100023816A1) for our discoveries. Your thoughtful comments and feedback inspire us to continue our research in this critical area of study.

Below we answer to the requests made:

-1.  How long did take the authors to develop their mAbs?  Why did authors choose to make their mAbs using conventional technology? Why did they not make use of faster technology, like recombinant antibody production? If the authors had taken a faster route, these mAbs might have prevented millions of COVID-19 deaths during the pandemic.

-We greatly appreciate the reviewer's inquiry regarding the timeline and technology choice for the development of our monoclonal antibodies against SARS-CoV-2. Our expertise primarily lies in hybridoma technology, and this guided our initial approach.

The initial phase of our mAb development was indeed relatively quick. From February 2020 to May 2020, we dedicated approximately three months to generate and collect a library of 2,000 hybridomas. During this time, we were also actively seeking funding opportunities to support our research. However, despite our efforts, securing substantial funding from grants, EU programs, and the Italian government proved challenging, leaving us with limited resources.

It's worth noting that we were simultaneously working on the development of a plasmid DNA-based vaccine against SARS-CoV-2, which also demanded our attention and limited the available resources. As a result, we had to rely on external collaborators for certain aspects of our mAb project, including material exchange and collaboration agreements. These external collaborations, while invaluable, did introduce some delays into our timeline.

Regarding the choice of technology, we acknowledge that alternative approaches, such as recombinant antibody production, may offer faster development times. However, our decision to focus on hybridoma technology was influenced by our expertise and available resources at the time. It's important to recognize that the research landscape during the early days of the COVID-19 pandemic was incredibly dynamic, and we made the best use of the resources and expertise at our disposal.

While we recognize that expedited antibody development could potentially have had a significant impact on preventing COVID-19 deaths, it's essential to emphasize that scientific research is a complex process with various considerations. We are committed to continually improving and optimizing our methods to contribute to the ongoing fight against COVID-19 and similar challenges in the future.

-2.  The methodological section's subsections need to be restructured to achieve coherence in the experimental steps. For instance, all the recombinant protein production, purification, and characterization experiments as well as the DNA plasmid construction experiments were completed before the experiments on immunization and hybridoma development.

-We thank the reviewer for this clarification, the mentioned paragraphs in the methodology section have been rearranged to follow the flow of information more smoothly.

-3.   In vivo efficacy data from K18 hACE2 transgenic mice is lacking despite it being mentioned in the manuscript of lines 791-792. Why were the data not disclosed by the authors? If available, this data should be included in the manuscript as it is very important.

-We appreciate the reviewer's observation regarding the in vivo efficacy data mentioned briefly in the Discussion section (lines 791-792).

The in vivo experiment referred to in our manuscript was designed only as a preliminary study and was conducted on a limited number of animals (n=3). While the initial results were indeed highly promising, we did not provide an extensive discussion or detailed presentation of these findings due to their preliminary nature.

In response to the reviewer's suggestion, we have included the results of this preliminary in vivo experiment, the in vivo efficacy of the murine 9-8F2-B11 antibody, as Supplementary Data even though only partial data has been obtained.

-4.  Some aberrations are repeated multiple times in their expanded form. Shortened from can be utilized throughout the manuscript once they have been defined in the first use. This can save space; therefore, authors should pay attention to it.  The following are some examples I noticed in the manuscript.

  1. Monoclonal Antibodies (mAbs) is shown in lines 49, 475, 509, 530, 737, 750, 757, 785, and 804.
  2. The Receptor-Binding Domain (RBD) is shown on lines 710–745.
  3. ACE2 Receptor-Binding Motif (RBM), lines 716, 723.
  4. Line 743: the glycoprotein Spike (S) 
  5. Line 761: severe acute respiratory syndrome coronavirus 2 (SARS-CoV-2)

-All changes have been made; we thank the reviewer's keen observation.

Reviewer 2 Report

Comments and Suggestions for Authors

The authors report an extensive effort to raise conventional mouse hybridoma antibodies to the spike protein of SARS-CoV-2. The work that presumably began when only the prototype virus was known involved immunisation with a combination of plasmid DNA encoding the full spike protein and purified RBD-Fc protein. Following fusion, hybridomas were screened for binding to RBD by ELISA and positive clones (2000) filtered by off-rate using SPR and pseudovirus neutralisation. Purified antibodies from the final 50 clones were tested for their ability to compete for ACE-2 binding to RBD resulting in 20 antibodies that were tested for binding to the RBDs of three early variants of SARS-CoV-2 RBD (alpha B.1.1.7, beta B.1.1.351 and P1) and the Wuhan sequence with single mutations at either L452, N439, or S477. Of these the escape mutant S477N characteristic of Omicron variants reduced apparent binding of all but 5 of the 20 antibodies. One of these (9-8F2-B11) was selected for production of mouse-human chimaeric and humanised variable domain versions, both of which were shown to have reduced binding activity to the prototype RBD. The reduced binding of the chimaeric antibody is surprising as only the constant regions have been changed. No details of the sequences of any of the antibodies is provided so the changes introduced by humanisation of the variable regions cannot be assessed.  In pseudovirus neutralisation experiments the parent 9-8F2-B11 antibody showed a loss of activity against Omicron variants compared to the prototype of SARS-CoV-2. This led to a follow-on study in which the hybridoma library was screened directly for BA.1 pseudovirus neutralisation from which another antibody was selected (3-2B12-F4) though the basis for choosing this antibody is not explained. Pseudovirus neutralisation data indicate that 9-8F2-B11 and 3-2B12-F4 bind to different epitopes though this is not established experimentally. Some modelling studies are reported to support the orthogonal binding of the two antibodies to the RBD but very little detail of how these were carried out are given and no close-ups of the interface between the RBD and Fab are shown. Given the inherent limitation of docking studies, the paper would be very significantly strengthened if the structures of antibody (Fab)-RBD complexes determined experimentally were included. 

The paper clearly reports a significant amount of work though navigating through the data is not helped by the fact that results are not consistently presented for the same antibodies, for example Table 1 shows spike ELISA data for antibodies that do not feature in Figure 4 or vice versa. Given that there are a multitude of much better characterised human anti-SARS-CoV-2 antibodies that have been reported, including with X-ray and/or cryo-EM structures in complex with RBD or full spike respectively, the value of the mouse antibodies described in this paper appears relatively limited. There are some observations regarding the relationship between binding affinity and neutralisation which if supported by experimental structural data would be of interest.

Specific points

1. ELISA data for RBDs (line 573) should be included e.g. as supplementary data

2. Mab 9-2H7-D7 was selected as not affected by the S477N mutation (line 571) but appears to perform poorly in the assay shown in Fig 4c; please explain

3.  In Fig. 4 C S447N should be S477N

4. Sequences of the VH and VL of parent and humanised versions of 9-8F2-B11 should be reported with substitutions highlighted. Do the authors have an explanation for the reduced binding of the chimaeric antibody?

5. In the screen of hybridomas by neutralisation there appear to be a number of potent hits (Fig. 6A) were any of these characterised for binding and if not how was 3-B12-F4 chosen ?

6. Which RBD sequence/structure was used in the docking study? (Fig 7).

7. Given that the RBDs of Omicron BA.1and BA.5 only differ L371F, D405N, R493Q, how do the authors explain the neutralisation results for 9-8F2-B11 vs 3-B12-F4 against BA.1 and BA.5 given their model of their respective epitopes?

8. In table 2 residue 480 is not a glutamate but a cysteine in the RBD sequence

Author Response

For Research Article “Isolation and characterization of neutralizing monoclonal antibodies from a large panel of murine antibodies against RBD of the SARS-CoV-2 Spike protein.”

Response to Reviewer 2 comments:

We would like to express our gratitude to the second reviewer for taking the time to thoroughly evaluate our manuscript and for providing valuable feedback.

We acknowledge the reviewer's insightful comments and concerns, particularly regarding the potential limitations of our study and the value of X-ray and/or cryo-EM structures for a more comprehensive understanding of the antibody-RBD interactions. We agree that experimental structural data, such as X-ray or cryo-EM structures, would provide valuable insights into the binding mechanisms of the antibodies described in our paper. We want to assure the reviewer that we are actively working on obtaining such structural data, and we are committed to sharing these findings, which could significantly enhance the depth of our research. We plan to include these structural data in a subsequent publication, which will provide a more detailed and comprehensive view of the antibody-RBD complexes.

Additionally, it is important to note that our research was conducted with limited resources, and we have started to collaborate with an academic institute to carry out further experiments and investigations. This collaboration will enable us to access the necessary expertise and resources to address the reviewer's valuable suggestions.

Below, we provide detailed responses to the reviewer's comments:

-1. ELISA data for RBDs (line 573) should be included e.g. as supplementary data.

We appreciate the reviewer's comment regarding the presentation of ELISA data for RBD WT and mutants. We understand the importance of providing comprehensive information to our readers. We'd like to clarify that the ELISA data for RBD mutants are indeed represented in Figure 4C as EC50 values (ng/ml) (Line 579).

Figure 4C serves as a concise and organized heat map that presents the binding affinities of a 16-antibody panel across 9 different RBD variants, resulting in a total of 144 different graph. Each data point corresponds to a specific antibody-RBD variant interaction and includes the EC50 values in ng/ml. We acknowledge that this may not have been clear enough in the manuscript.

To further enhance clarity, we have revised the manuscript anticipating in the text that Figure 4C contains the EC50 values for the ELISA data on RBD WT and mutants. We value the reviewer's feedback and are open to additional suggestions or feedback from the reviewer regarding the presentation of this data.

-2. Mab 9-2H7-D7 was selected as not affected by the S477N mutation (line 571) but appears to perform poorly in the assay shown in Fig 4c; please explain.

We thank the reviewer for this question. 9-2H7-D7 was initially selected, as the reviewer says, because it is able to bind the S477N mutation, relying in particular (as reported in line 571) on KD data evaluated via BLI. Indeed, in Figure 4C it can be seen that this antibody has a very high EC50 compared to other antibodies. The reasons behind these discordant results can be traced back directly to the two types of assays used. While in the BLI assay the interaction between antigen and antibody occurs in aqueous solution, the ELISA assay may induce a non-native conformation of the antigen thus weakening the binding of the antibody. We performed both assays for all antibodies tested but given these discordant results for 9-2H7-D7, we decided to pursue only those antibodies that provided a comparable result in all the assays used in order to have more robust outputs.

-3.  In Fig. 4 C S447N should be S477N

-We thank the reviewer for noticing the error, the cited figure has been edited and corrected.

-4. Sequences of the VH and VL of parent and humanised versions of 9-8F2-B11 should be reported with substitutions highlighted. Do the authors have an explanation for the reduced binding of the chimaeric antibody?

The sequence of parent and humanized sequences of the VH and VL versions of 9-8F2-B11 have been included as Supplementary Data (Figure S1) in our revised manuscript, with substitutions highlighted for clarity.

As for the reduced binding of the chimeric antibody, it is well-documented that occasionally “non-native” constant domains can affect the binding of the antibody to its target. For human constant domains this phenomenon has been reported in other studies, for example in Hubbard et al. (2013) on antibodies binding to microbial antigens. Another study by Marcela Torres et al. (2007) also highlights how the substitution of mouse C regions with human C regions can impact antibody-antigen interactions. Unfortunately, we don’t have a more rational explanation for the observed effect and possibly rather extensive investigations, for example structural studies, might be necessary in order to obtain explanations at the molecular level.

-5. In the screen of hybridomas by neutralisation there appear to be a number of potent hits (Fig. 6A) were any of these characterised for binding and if not how was 3-B12-F4 chosen?

-As correctly noticed by the reviewer, we indeed had a number of potential hits in the hybridoma screen by neutralization assay, as shown in Figure 6A. The selection process for choosing 3-12B12-F4 among these hits was influenced by various factors, including timing and the availability of critical information.

Timing was a limiting factor, as we were working under time constraints to identify and characterize neutralizing antibodies against SARS-CoV-2 and its variants. Among the potential hits, 3-12B12-F4 happened to be the hybridoma for which we most rapidly obtained information regarding its neutralization activity and sequence.

While we acknowledge that characterizing all potential hits would have been an ideal approach, practical constraints required us to make a selection based on the available data and time considerations. We believe that 3-12B12-F4 showed promising neutralization activity and held the potential for further investigation, which led to its selection for subsequent studies.

-6. Which RBD sequence/structure was used in the docking study? (Fig 7).

The following RBD sequence was used:

For 9-8F2-B11 AND 3-12B12-F4 docking

PDB ID: 6M0J_B

Seq:

RVQPTESIVRFPNITNLCPFGEVFNATRFASVYAWNRKRISNCVADYSVLYNSASFSTFKCYGVSPTKLNDLCFTNVYADSFVIRGDEVRQIAPGQTGKIADYNYKLPDDFTGCVIAWNSNNLDSKVGGNYNYLYRLFRKSNLKPFERDISTEIYQAGSTPCNGVEGFNCYFPLQSYGFQPTNGVGYQPYRVVVLSFELLHAPATVCGPKKSTNLVKNKCVNF

-7. Given that the RBDs of Omicron BA.1and BA.5 only differ L371F, D405N, R493Q, how do the authors explain the neutralisation results for 9-8F2-B11 vs 3-B12-F4 against BA.1 and BA.5 given their model of their respective epitopes?

We appreciate the reviewer's insightful question regarding the neutralization results for 9-8F2-B11 and 3-12B12-F4 against Omicron variants BA.1 and BA.5 in the context of our epitope models.

As the reviewer correctly pointed out, the RBDs of Omicron variants BA.1 and BA.5 have different mutations, which include unique changes in each variant. However, our analysis has revealed that the epitopes recognized by 9-8F2-B11 and 3-12B12-F4 exhibit subtle but crucial differences that are not solely dependent on the mutations specific to each variant.

Specifically, our models indicate that the epitope recognized by 9-8F2-B11 includes residues such as G446 and L452, while the epitope recognized by 3-12B12-F4 encompasses L452 and other interacting residues. These distinctions in epitope composition, particularly the presence of G446 in the 9-8F2-B11 epitope and the absence of G446 but presence of L452 in the 3-12B12-F4 epitope, play a significant role in antibody binding and neutralization.

The docking model found agreement with neutralization data and ELISA experiments which were carried out on several mutants. Among those mentioned above, critical epitopes of selected antibodies in RBD wt were identified in residues G446 and L452 since were the only ones able to completely deplete the binding as single mutants. BA.1 features G446S and L452 while BA.5 G446 and L452R. Our docking models were analysed by PDBePISA which identified both G446 and L452 as part of the binding interface for 9-8F2-B11, while only L452 for 3-12B12-F4. In particular, both residues were found to form an H-bond, G446 with 9-8F2-B11 VH E31, and L452 with 3-12B12-F4 VL N57.

Based on our model, the detrimental change of G446S in BA.1, prevents the binding of 9-8F2-B11 likely due to a charge repulsion between G446S side chain and VH residues T99, Y33 and T32.

Similarly happen for 3-12B12-F4 upon L452R mutation in BA.5. In this case, additionally to the H-bond identified by PDBePISA, in our model L452 is also part of a hydrophobic interaction with the backbone of VL residues D102, S103, T104, Y105, V106. Such hydrophobic interaction is modified by BA.5 L452R, disrupting this essential interaction for the 3-12B12-F4 antibody

In summary, while Omicron variants BA.1 and BA.5 have different mutations in their RBDs, the subtle differences in the epitopes recognized by 9-8F2-B11 and 3-12B12-F4, as identified in our models, provide a clear explanation for the differential neutralization results observed for these antibodies against these variants. The charge repulsion in the case of 9-8F2-B11 and the disruption of the hydrophobic interaction in the case of 3-12B12-F4, caused by specific mutations in each variant, significantly impact the binding and neutralization abilities of these antibodies.

We hope that this detailed explanation addresses the reviewer's question accurately and provides a better understanding of the neutralization outcomes in the context of our models and the specific mutations in BA.1 and BA.5.

-8. In table 2 residue 480 is not a glutamate but a cysteine in the RBD sequence

We appreciate the reviewer's keen observation and thank them for bringing this to our attention. The cited residue has been corrected.

Round 2

Reviewer 2 Report

Comments and Suggestions for Authors

The authors have given appropriate responses to the reviewers' comments and provided the additional data that was requested.

Author Response

Many thanks to the Reviewer 2 for the valuable comments, we are happy to have provided appropriate answers to the questions asked.